# 27-Hydroxycholesterol Negatively Affects the Function of Bone Marrow Endothelial Cells in the Bone Marrow

**DOI:** 10.3390/ijms251910517

**Published:** 2024-09-29

**Authors:** Soo-Yeon Woo, Wan-Seog Shim, Hyejin Lee, Ninib Baryawno, Parkyong Song, Byoung Soo Kim, Sik Yoon, Sae-Ock Oh, Dongjun Lee

**Affiliations:** 1Department of Convergence Medicine, School of Medicine, Pusan National University, Yangsan 50612, Republic of Korea; woolovein@naver.com (S.-Y.W.); sws4362@gmail.com (W.-S.S.); 4247a@naver.com (H.L.); parkyong.song@pusan.ac.kr (P.S.); 2Childhood Cancer Research Unit, Department of Women’s and Children’s Health, Karolinska Institutet, 17177 Stockholm, Sweden; n.baryawno@ki.se; 3School of Biomedical Convergence Engineering, Pusan National University, Yangsan 50612, Republic of Korea; bskim7@pusan.ac.kr; 4Department of Anatomy, School of Medicine, Pusan National University, Yangsan 50612, Republic of Korea; sikyoon@pusan.ac.kr (S.Y.); hedgehog@pusan.ac.kr (S.-O.O.); 5Transplantation Research Center, Research Institute for Convergence of Biomedical Science and Technology, Pusan National University Yangsan Hospital, Yangsan 50612, Republic of Korea

**Keywords:** 27-hydroxycholesterol, bone marrow endothelial cell, hematopoietic stem cell

## Abstract

Hematopoietic stem cells (HSCs) reside in specific microenvironments that facilitate their regulation through both internal mechanisms and external cues. Bone marrow endothelial cells (BMECs), which are found in one of these microenvironments, play a vital role in controlling the self-renewal and differentiation of HSCs during hematological stress. We previously showed that 27-hydroxycholesterol (27HC) administration of exogenous 27HC negatively affected the population of HSCs and progenitor cells by increasing the reactive oxygen species levels in the bone marrow. However, the effect of 27HC on BMECs is unclear. To determine the function of 27HC in BMECs, we employed magnetic-activated cell sorting to isolate CD31^+^ BMECs and CD31^−^ cells. We demonstrated the effect of 27HC on CD31^+^ BMECs and HSCs. Treatment with exogenous 27HC led to a decrease in the number of BMECs and reduced the expression of adhesion molecules that are crucial for maintaining HSCs. Our results demonstrate that BMECs are sensitively affected by 27HC and are crucial for HSC survival.

## 1. Introduction

The bone marrow (BM) plays a crucial role in the production of blood cells and consists of a diverse range of cell types, including both blood- and non-blood-forming cells. Hematopoiesis is regulated by hematopoietic stem cells (HSCs), which regenerate many blood cell types through multipotent differentiation [1]. In addition, HSCs possess the ability to regenerate, thereby ensuring the maintenance of HSC homeostasis throughout an individual’s lifetime [2]. HSCs are located in specific microenvironments known as “niches”, which utilize both internal mechanisms and external cues to control the features of HSCs [3]. These niches consist of many cell types including osteoblasts, mesenchymal stromal cells, adipocytes, macrophages, neuronal cells, endothelial cells (ECs), and perivascular cells [4]. These are connected in terms of their physical structures and functionalities. HSCs are situated near ECs and interact with adjacent cells in specialized microenvironments [5].

Bone marrow ECs (BMECs) organize vascular niches and produce cytokines required for the self-renewal and differentiation of HSCs [6]. BMECs predominantly release stem cell factor (SCF) and CXC motif chemokine 12 (CXCL12), which are crucial for the maintenance of HSCs [7,8]. SCF, encoded by the kit ligand gene (*Kitl*), is an essential growth factor for HSC maintenance [9]. ECs are a heterogeneous cell population comprising arteriolar BMECs and sinusoidal BMECs [10]. Arteriole BMECs and sinusoidal BMECs influence HSC maintenance and regeneration through the differential expression of SCF and CXCL12 [11,12]. In addition, capillary endothelial cells release apelin, which stimulates hematopoietic reconstitution following myeloablative injury [13]. BMECs contain a wide range of adhesion molecules, including VCAM1 and ICAM1, which interact with LFA1 and VLA4 on HSCs [14]. This interaction modulates the capability of HSC retention or leads to alteration in BMEC-derived cytokine production [15,16].

In the steady state, HSCs remain quiescent but are activated by endogenous and exogenous stimulation induced by various stresses [17]. Nakada et al. found that female mice exhibited a higher rate of HSC division compared with that in male mice. This disparity is due to the presence of estrogen, a primary sex hormone mostly produced in the ovaries [18]. In addition, estrogen increases extramedullary hematopoiesis by promoting HSC mobilization during pregnancy [19]. Estrogen receptor α (ERα) is expressed in HSCs. Additionally, 27-hydroxycholesterol (27HC), an oxysterol, acts as a selective estrogen receptor modulator (SERM) and binds to ERα [20]. Like estrogen, 27HC is involved in HSC mobilization from the BM to the spleen in an Erα-dependent manner during pregnancy [19].

27HC is a cholesterol metabolite produced by sterol hydroxylase CYP27A1 and metabolized by oxysterol hydroxylase CYP7B1 [21]. In the blood of healthy adults, 27HC is the most abundant oxysterol [22]. 27HC acts as a ligand for the liver X receptor (LXR) and regulates lipid homeostasis [23]. Furthermore, 27HC affects cancer development and metastasis via a proinflammatory process mediated by ERα in the vasculature [24,25,26,27]. A recent study demonstrated that hypercholesterolemia-induced endothelial activation and macrophage-derived 27HC promote atherosclerosis through cross-talk with endothelial ERα [28].

Due to the fact that 27HC reduces the hematopoietic stem and progenitor cell (HSPC) population by increasing reactive oxygen species (ROS) [29], we examined whether 27HC is also important for the function of BMECs using mouse bone marrow cells. Furthermore, we investigated how the treatment of BMECs with exogenous 27HC affects the lifespan of HSCs, given that BMECs affect HSC survival [30]. In this study, to determine the function of 27HC in BMECs, we isolated BMECs and HSCs based on CD31 (PECAM1) expression, which is known to act as an endothelial surface marker. By doing so, we uncover the influence of 27HC in BMECs, including alterations in BMEC adhesion molecules and BMEC-derived cytokine that regulate HSC function. Collectively, our data demonstrate that BMECs are sensitive to the effects of 27HC and that the presence of BMECs is crucial for the survival of HSCs.

## 2. Results

### 2.1. 27HC Has a Greater Impact on BMECs than on HSCs

27HC was previously shown to decrease the immature HSPC population from isolated mouse bone marrow cells [29]. However, the effects of 27HC on BMECs remain unclear. HSCs interact with BMECs to maintain homeostasis [31]. As previous studies revealed that 27HC leads to ROS generation, endoplasmic reticulum stress, and apoptosis in HSPCs [29], we examined the effect of 27HC on BMECs. We first examined the frequency of BMECs after 27HC treatment. The exogenous 27HC treatment of BM cells led to a decrease in the numbers of total ECs (CD31^+^Ter119^−^CD45^−^), sinusoidal BMECs (sBMECs; CD31^+^Ter119^−^CD45^−^Sca1^−^), and arterial BMECs (aBMECs; CD31^+^Ter119^−^CD45^−^Sca1^+^) (Figure 1). Additionally, the EC population was significantly reduced even with low-dose 27HC treatment (Figure 2A,B), whereas the HSC population was no different in its frequencies (CD31^+^Ter119^−^CD45^−^Sca1^+^) after the 27HC treatment for 48 h (Figure 2C,D).

We previously observed that intraperitoneal administration of 27HC did not affect the HSPC population in a steady state [29]. To test whether exogenous 27HC affects the BMEC population in vivo, we treated mice with 27HC (5 mg/kg/d) for 4 weeks and analyzed the immature HSC and BMEC populations in the BM (Figure 3A). As 27HC injection did not significantly reduce the HSC population [29], we then conducted 27HC long-term intraperitoneal injection. As we described previously [29], long-term 27HC administration did not affect the number of LKS (Lin^−^Sca1^+^cKit^+^), hematopoietic progenitor cells (HPCs; Lin^−^Sca1^+^cKit^+^CD48*), and HSCs (Figure 3B), but significantly decreased the BMEC population (Figure 3C). These results suggest that exogenous 27HC treatment has stronger effects on BMECs than on HSCs.

### 2.2. CD31^+^ BMECs but Not HSCs Are Impacted by 27HC

CD31 (PECAM-1) is known to highly express endothelial cells [32]. To further explore the specific effects of 27HC on BMECs and HSCs, we tried to isolate BMECs and HSCs based on CD31 expression from mouse bone marrow cells. First, we used magnetic-activated cell sorting (MACS) to isolate CD31^+^ BMEC cells and CD31^−^ non-EC cells (Figure 4A). CD31^+^ cells and CD31^−^ cells were plated on cell culture plates and treated with 27HC for 48 h. After that, we analyzed the frequency of BMECs from CD31^+^ cells and of HSCs from CD31^−^ cells. The BMEC population decreased after 27HC treatment according to the results of a WST1 assay and fluorescence-activated cell sorting (FACS) analysis (Figure 4B,C). We also measured the HSC population from CD31^−^ cells. The results show that HSCs were not affected by exogenous 27HC treatment (Figure 4D). These results demonstrate that low-dose 27HC differentially affects BMECs and HSCs.

27HC was previously shown to induce apoptosis in HSPCs through the accumulation of ROS, which activated the ER stress response [29]. ER stress was causative of cell death through several mechanisms [33]. To confirm the causes of decreased BMEC after 27HC exposure, we assessed the ROS level by measuring 2′,7′-dichlorofluorescein diacetate (DCFDA). The results show that 27HC treatment causes increased ROS levels in BMECs. In addition, we assessed the ER stress pathway in 27HC-treated BMECs. 27HC treatment augments the expression of phospho-eukaryotic initiation factor 2 (peIF2a) involved in the ER stress pathway in BMECs upon 27HC treatment. Collectively, these data suggest that 27HC induces a reduction in the BMEC population through the ER stress response pathway activation by the accumulation of ROS (Figure 5).

BMECs are an essential component of the perivascular niche for HSC maintenance and express various factors to regulate HSC functions [34]. BMECs express adhesion molecules such as ICAM1 and VCAM1. These adhesion molecules interact with HSC adhesion receptors to maintain the quiescent state of HSCs [35]. According to Liu et al., ICAM1 deficiency in the bone marrow niche results in the dysregulation of HSC quiescence, self-renewal, and regeneration capacity [14]. Furthermore, VCAM1 deficiency in BMECs leads to increased HSC mobilization and the loss of self-renewal ability [36]. Consistent with these observations, flow cytometric analysis showed that exogenous 27HC treatment decreased ICAM1 and VCAM1 expression in BMECs (Figure 6A,B).

In addition, BMECs modulated the homeostasis of HSCs through the secretion of specific angiocrine factors [37]. NOTCH signaling maintains HSCs in an immature state and regulates the HSC self-renewal function [38,39]. Expression of NOTCH1 was previously shown to increase in the reconstitution of HSCs after myeloablation [40]. In addition, NOTCH1 receptor expression and activation in BMECs were increased after 5-fluorouracil (5-FU) treatment that induced myelosuppression [41]. Consistent with these observations, 27HC treatment increased NOTCH1 expression level in BMECs (Figure 6C). These data suggest that damaged BMECs by 27HC promote NOTCH1 expression to support regeneration.

TNFα is a cytokine exerting both inhibitory and stimulatory effects on a diversity of cellular processes [42]. TNFα secreted from the BMECs is well-known to affect HSC growth and maintenance. Overexpression of TNFα could contribute to several bone marrow failure syndromes such as Fanconi anemia [43]. Thus, these findings directed us to explore the *TNFα* level post-27HC treatment in BMECs. Our findings reveal that 27HC treatment augments the expression of *Tnfα* mRNA level in the BMECs (Figure 6D).

### 2.3. BMEC Dysfunction by Exogenous 27HC Treatment Affects HSCs

Since 27HC induces BMEC dysfunction, we next sought to determine how BMEC-secreted factors by 27HC impact the HSCs. We assessed the impact of CD31^+^ BMEC conditioned medium on the function of HSCs. CD31^+^ cells isolated by MACS were treated with 27HC for 48 h and then changed to a growth medium without 27HC and incubated for 24 h. The reason for changing the medium without 27HC was to ablate the remaining 27HC influence in whole bone marrow cultures. After 24 h, this conditioned medium was collected and used to treat mouse whole bone marrow cells for 48 h. When mouse whole bone marrow cells were cultured in a 27HC-treated CD31^+^ BMEC conditioned medium, the frequency of LKS, HPCs, and HSCs decreased. Taken together, these data suggest that 27HC potently induces the dysfunction of BMECs which are important for HSC survival (Figure 7).

## 3. Discussion

We previously demonstrated that 27HC reduces the HSPC population by increasing ROS levels and the endoplasmic reticulum stress response [29]. In this study, we examined the effects of 27HC on BMECs. Treatment with exogenous 27HC led to a considerable dose-dependent decrease in the number of BMECs. In addition, 27HC potently induces the dysfunction of BMECs that are important for HSC survival.

Cholesterol is an essential component of the mammalian cell membrane and is a precursor of bile acids and steroid hormones [29]. Cholesterol plays an important role in cellular function but is also involved in cancer [44]. In addition, hypercholesterolemia promotes the mobilization of HSPCs and endothelial activation [45,46]. Oxysterol, an oxidative derivative of cholesterol, is involved in malignancies [47]. 27HC is the primary metabolite of cholesterol and the most abundant circulating oxysterol in the blood of healthy adults. 27HC levels are correlated with hypercholesterolemia and are elevated within atherosclerotic plaques and tumor tissues [27]. Additionally, 27HC functions as an SERM and LXR agonist, thereby increasing ERα-dependent tumor growth and LXR-dependent metastasis [48]. Hypercholesterolemia influences hematopoiesis [29,49,50]. Tie et al. reported that hypercholesterolemia promotes the aging of HSCs and impairs their reconstitution capacity by inducing oxidative stress [49]. We previously demonstrated that exogenous 27HC induces a decrease in the HSPC population by increasing ROS levels, the endoplasmic reticulum stress response, and apoptosis in HSPCs [29]. In addition, we verified the therapeutic effects of 27HC on human leukemia cells. As HSC self-renewal and differentiation are controlled by interacting signals from the BM niche [50], we hypothesized that 27HC also affects other cells in the BM niche.

Blood vessels are extensively distributed throughout the BM and are connected to perivascular cells to organize distinct niches for HSC regulation [51]. ECs organize the inner walls of blood vessels and are essential to maintain the barrier function of the vasculature [52]. ICAM1 and VCAM1 are adhesion molecules that are mostly expressed in ECs [53]. Adhesion molecules facilitate the attachment of HSCs to the vascular niche, thereby controlling their function [54]. NOTCH signaling is also involved in EC and HSC interactions [55]. ECs express Jagged1 and Jagged 2, and HSCs express NOTCH2. NOTCH signaling promotes HSC regeneration after myelosuppression via the direct binding of ligands and receptors [55]. Furthermore, ECs secrete growth factors, such as SCF, CXCL12, and pleiotrophin, which are important for the maintenance and regeneration of HSCs [7,8,12]. Poulous et al. demonstrated that the transplantation of BMECs into lethally irradiated mice improves hematopoietic recovery [30]. These data suggest that ECs are indispensable for the maintenance and regeneration of HSCs.

## 4. Materials and Methods

### 4.1. Animals

All animal experiments were approved by the Pusan National University School of Medicine. Experiments were performed using 8–12-week-old C57BL/6J wild-type mice. C57BL/6J wild-type mice were purchased from KOATECH (Seoul, Republic of Korea).

### 4.2. Mouse BM Cell Culture and 27HC Treatment

BM cells were extracted from the femurs and tibias of the mice. To extract mouse BM cells, the bone ends were sliced off and the bones were flushed with FACS buffer containing phosphate-buffered saline, 2% fetal bovine serum, and 0.8% penicillin/streptomycin. The dissociated BM cells were treated with ACK lysis buffer to dissolve erythrocytes for 10 min at 20–25 °C. Mouse BM cells (approx. 3 × 10^7^ cells) were cultured in RPMI medium supplemented with 2 mM glutamine, 10% fetal bovine serum, 1% penicillin/streptomycin, IL3, IL6, and SCF. 27HC was dissolved in ethanol and added to the growth medium [29]. The control group was administered an equal volume of ethanol.

### 4.3. Flow Cytometric Analysis

Flow cytometry was performed using antibodies as previously described [10,29,56]. To analyze the frequency of SLAM and endothelial cells, BM cells were stained with fluorescence-conjugated antibodies (Table 1). Intracellular ROS levels were assessed by 2′,7′-dichlorofluorescin diacetate (DCFDA; Invitrogen, Carlsbad, CA, USA, Thermo Fisher Scientific, Waltham, MA, USA) for 30 min at 37 °C. Intracellular phospho-protein staining was performed as previously described [57]. Briefly, cells were incubated with primary anti-peIF2 and NOTCH antibodies (Cell Signaling Technologies, Danvers, MA, USA) in FACS buffer, which consisted of phosphate-buffered saline (PBS) containing 2% FBS for 30 min at 4 °C. The cells were washed and incubated with Alexa Fluor 488-conjugated secondary antibody (Invitrogen) for 30 min at 4 °C. After washing with FACS buffer, flow cytometry analysis was performed on a FACSCanto 2 flow cytometer (BD Biosciences, Franklin Lakes, NJ, USA) using FlowJo software version 10 (TreeStar, Ashton, OR, USA).

### 4.4. WST-1 Assay

For the WST-1 assay, CD31^+^ cells were seeded into 96-well culture plates at a density of 1 × 10^5^ cells per well. The cells were exposed to various concentrations of 27HC for 48 h. WST-1 reagent (10 µL) was added to each well. The cells were incubated for an additional 1 h in a CO_2_ incubator at 37 °C. The absorbance of the solution was measured at 450 nm, with a reference wavelength of 620 nm. This procedure was performed according to the manufacturer’s instructions (Sigma Aldrich, St. Louis, MO, USA).

### 4.5. Magnetic Cell Sorting Analysis

Magnetic cell sorting was performed as previously described [58]. To accomplish magnetic cell sorting, we used MACS cell separation (Miltenyi Biotech, Bergisch Gladbach, Germany). Up to 1 × 10^8^ mouse BM cells were labeled with CD31 microbeads. CD31^+^ cells were separated by passing CD31 microbead-labeled BM cells through LS columns.

### 4.6. Quantitative RT-PCR

Total RNA was extracted with the easy-BLUETM total RNA extraction kit (iNtRON biotechnology, Seongnam, Republic of Korea). RNA quantification was performed on a Nanodrop 2000 spectrophotometer (Thermo, Waltham, MA, USA). Subsequently, cDNA was synthesized using the High Capacity cDNA Archive Kit (Applied Biosystems, Waltham, MA, USA) according to the manufacturer’s protocols. Real-time PCR amplification was conducted on a QuantStudioTM Real-Time PCR System (Applied Biosystem, Waltham, MA, USA) using SYBRTM green PCR Master Mix (Applied Biosystems, Waltham, MA, USA). The cycling conditions were as follows: initial denaturation at 95 °C for 10 min, followed by 40 cycles of 95 °C for 15 s, 60 °C for 15 s, and 72 °C for 30 s. Post-amplification, a melting curve analysis was performed according to the manufacturer’s instructions.

### 4.7. Statistical Analysis

All data are expressed as the means ± SEM. The two-tailed unpaired Student’s *t*-test was used to calculate the significance of differences between two groups, and one-way ANOVA with Tukey’s multiple comparisons among groups (Table 2). GraphPad Prism 5 software (GraphPad, Inc., La Jolla, CA, USA) was used for data plotting and statistical analyses. The level of significance is represented as follows: * *p* < 0.05, ** *p* < 0.01, *** *p* < 0.001.

## 5. Conclusions

We demonstrated that exogenous 27HC treatment impairs BMEC adhesion molecules. HSC and BMEC interactions within the BM niche are shaped by adhesion. BMECs express adhesion molecules such as ICAM1 and VCAM1, which interact with HSC adhesion receptors to maintain cell function [59]. Recent studies reported that ICAM1 is required for the quiescence, maintenance, self-renewal, and differentiation of HSCs [14]. HSCs from ICAM1-deficient mice changed their location within the BM niche, which can induce HSC impairment [60]. Additionally, VCAM1 is important for modulating HSC self-renewal by retaining HSCs within the BM niche [16]. As exogenous 27HC treatment reduced the levels of BMEC adhesion molecules, our data demonstrate that BMECs are sensitive to 27HC and that these cells are crucial for the survival of HSCs.

## Figures and Tables

**Figure 1 ijms-25-10517-f001:**
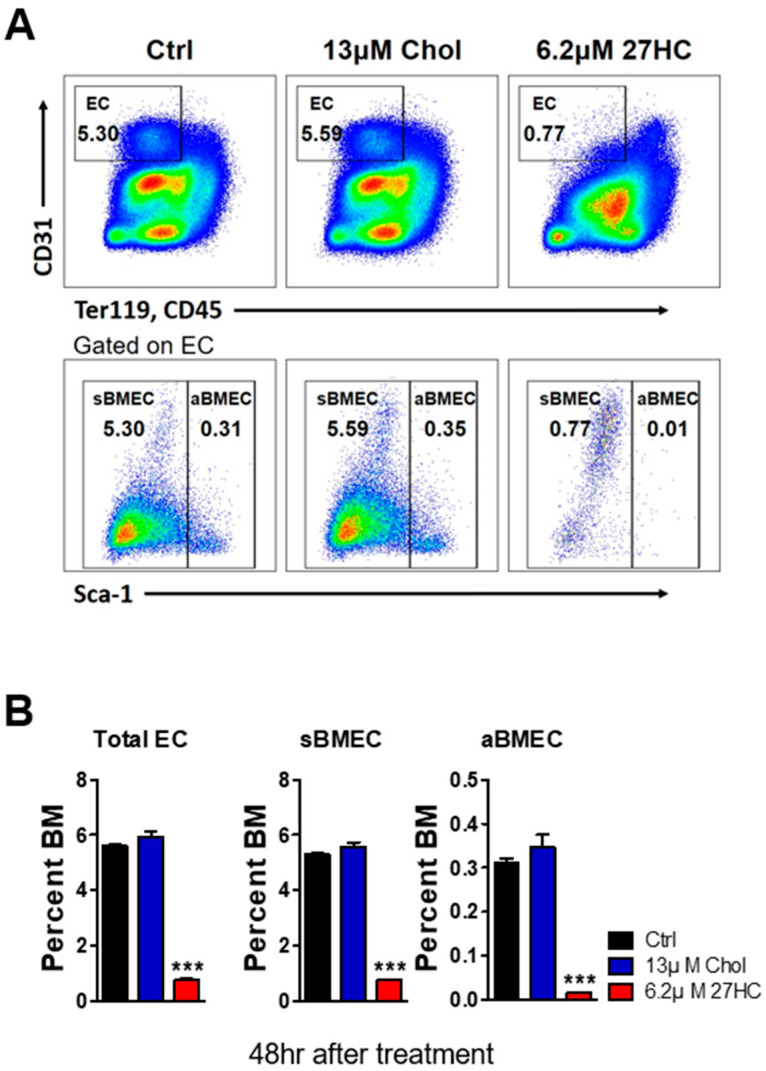
27-Hydroxycholesterol (27HC) impairs bone marrow endothelial cells (BMECs). Whole bone marrow cells (approx. 3 × 10^7^ cells) were seeded into 6-well plates and treated with ethanol (Ctrl), 13 µM cholesterol (13 µM Chol), and 6.2 µM 27-Hydroxyhcholesterol (6.2 µM 27HC) for 48 h at 37 °C. The frequency of endothelial cells was analyzed using fluorescence-activated cell sorting (FACS). (**A**) FACS plot showing the frequency of total endothelial cells (ECs; CD31^+^Ter119^−^CD45^−^), sinusoidal bone marrow endothelial cells (sBMECs; CD31^+^Ter119^−^CD45^−^Sca1^−^), and arteriole bone marrow endothelial cells (aBMECs; CD31^+^Ter119^−^CD45^−^Sca1^+^) after 27HC treatment. (**B**) The frequencies of total ECs, sBMECs, and aBMECs were decreased in the mouse bone marrow cells after 6.2 µM 27HC treatment. Data represent the mean ± SEM, one-way ANOVA followed by Tukey’s multiple comparisons test. *** *p* < 0.001.

**Figure 2 ijms-25-10517-f002:**
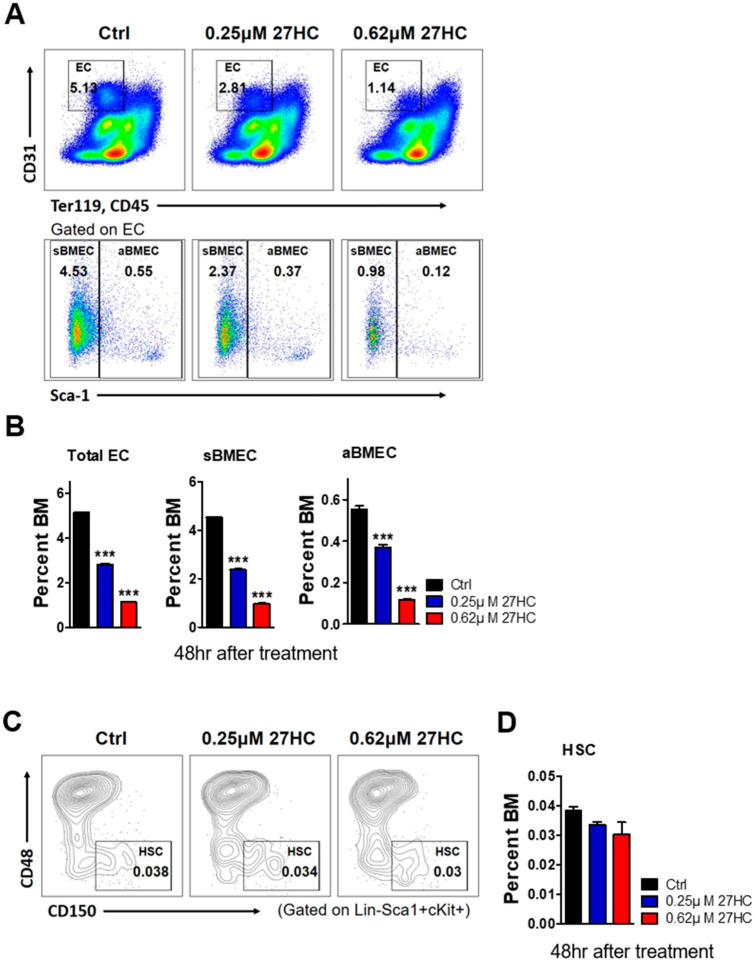
Low-dose 27HC affects endothelial cells but not hematopoietic stem cells (HSCs). Whole bone marrow cells were seeded into 6-well plates and treated with 27HC (0.25 µM and 0.62 µM) for 48 h at 37 °C. The frequency of endothelial cells and HSCs was analyzed using fluorescence-activated cell sorting (FACS). (**A**) FACS plot showing the frequency of total endothelial cells (ECs; CD31^+^Ter119^−^CD45^−^), sinusoidal bone marrow endothelial cells (sBMECs; CD31^+^Ter119^−^CD45^−^Sca1^−^), and arteriole bone marrow endothelial cells (aBMECs; CD31^+^Ter119^−^CD45^−^Sca1^+^) after 27HC treatment. (**B**) The frequencies of total ECs, sBMECs, and aBMECs were decreased in the mouse bone marrow cells after 27HC treatment. (**C**) FACS plot showing the frequency of hematopoietic stem cells (HSCs; Lin^−^Sca1^+^cKit^+^CD150^+^CD48^−^) after 27HC treatment. (**D**) The frequencies of HSCs were observed after 27HC treatment. Data represent the mean ± SEM, one-way ANOVA followed by Tukey’s multiple comparisons test. *** *p* < 0.001.

**Figure 3 ijms-25-10517-f003:**
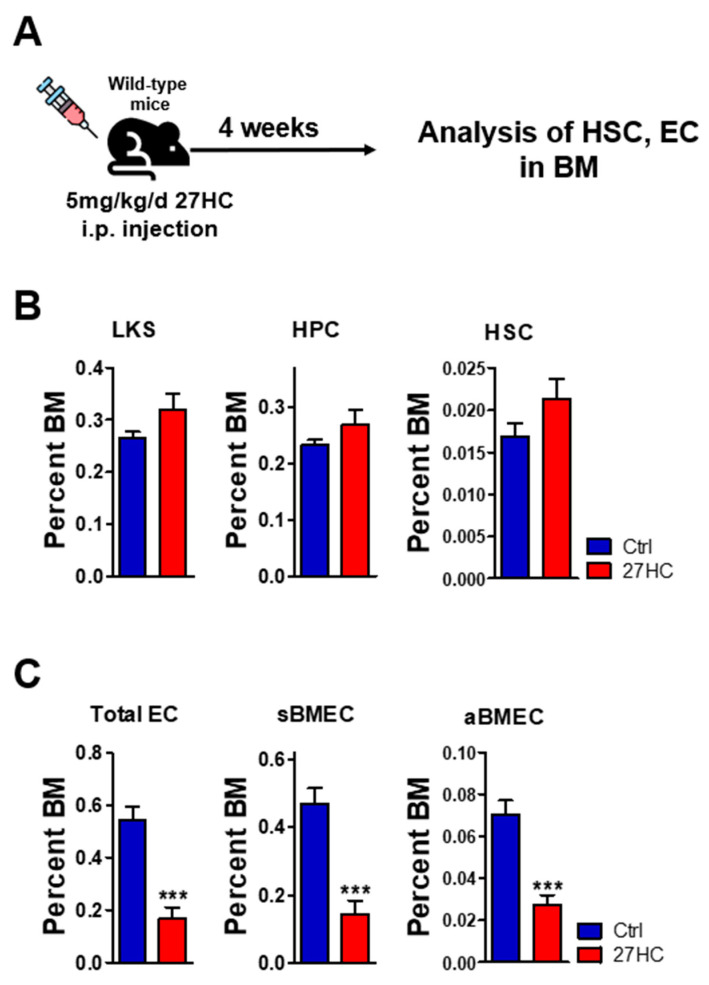
Injection of 27HC in vivo. C57BL/6 mice were injected daily with 27HC (5 mg/kg, n = 5) for 4 weeks. Whole bone marrow cells were harvested from the mice’s tibias and femurs. The frequencies of BMECs and HSCs were tested using flow cytometric analysis. (**A**) Scheme of experiments. (**B**) No differences in the frequencies of LKS (Lin^−^Sca1^+^cKit^+^), hematopoietic progenitor cells (HPCs; Lin^−^Sca1^+^cKit^+^CD48^+^), and HSCs (Lin^−^Sca1^+^cKit^+^CD150^+^CD48^−^) were observed after exogenous 27HC intraperitoneal injection. (**C**) Frequencies of total endothelial cells (ECs; CD31^+^Ter119^−^CD45^−^), sinusoidal bone marrow endothelial cells (sBMECs; CD31^+^Ter119^−^CD45^−^Sca1^−^), and arteriole bone marrow endothelial cells (aBMECs; CD31^+^Ter119^−^CD45^−^Sca1^+^) were significantly decreased by exogenous 27HC intraperitoneal injection. Data represent the mean ± SEM, unpaired *t*-test. *** *p* < 0.001.

**Figure 4 ijms-25-10517-f004:**
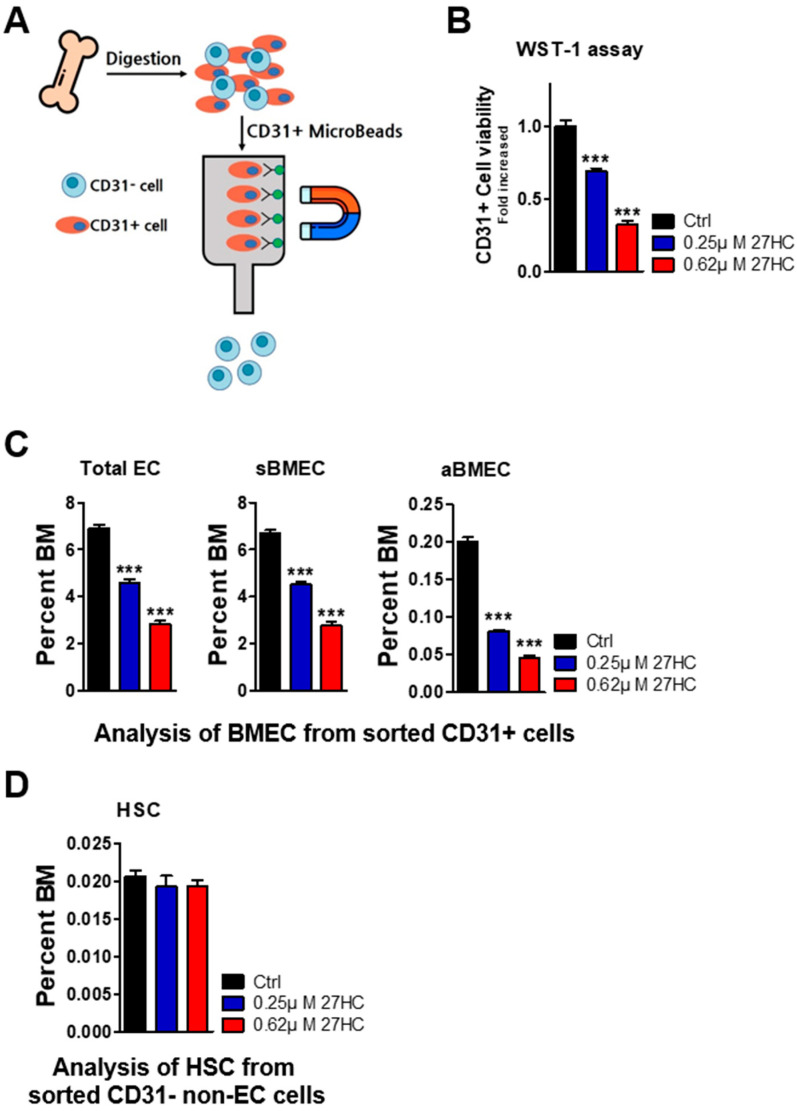
Low-dose 27-Hydroxycholesterol (27HC) affects CD31^+^ bone marrow endothelial cells (BMECs) but not hematopoietic stem cells (HSCs). CD31^+^ cells and CD31^−^ cells were sorted from mouse bone marrow cells using magnetic-activated cell sorting (MACS) cell separation. CD31^+^ cells were seeded into 6-well plates with rat tail collagen type I. CD31^+^ and CD31^−^ cells were treated with 27HC (0.25 and 0.62 µM) for 48 h. The frequencies of BMECs from CD31^+^ cells and of HSCs from CD31^−^ cells were analyzed using flow cytometric analysis. (**A**) Schematic representation of the experiments. (**B**) The viability of CD31^+^ cells was significantly decreased by 27HC treatment, as measured in a WST-1 assay. (**C**) Frequencies of total endothelial cells (ECs; CD31^+^Ter119^−^CD45^−^), sinusoidal bone marrow endothelial cells (sBMECs; CD31^+^Ter119^−^CD45^−^Sca1^−^), and arteriole bone marrow endothelial cells (aBMECs; CD31^+^Ter119^−^CD45^−^Sca1^+^) in CD31^+^ cells were significantly decreased by exogenous 27HC treatment. (**D**) The frequency of HSCs (Lin^−^Sca1^+^ c-Kit ^+^CD150^+^CD48^−^) among CD31^−^ cells did not differ. Data represent the mean ± SEM, one-way ANOVA followed by Tukey’s multiple comparisons test. *** *p* < 0.001.

**Figure 5 ijms-25-10517-f005:**
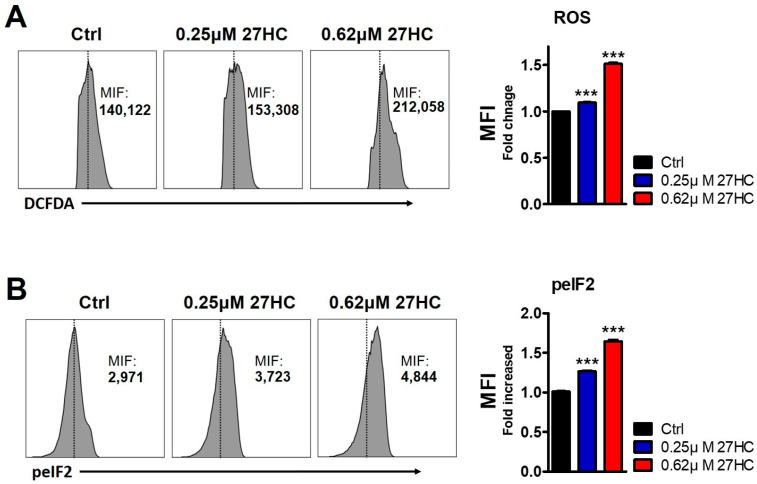
27HC increases reactive oxygen species (ROS) and endoplasmic reticulum (ER) stress in BMECs. CD31^+^ cells are treated with 27HC (0.25 and 0.62 µM) for 48 h. (**A**) FACS plot showing the DCFDA expression level of the BMEC population after 27HC treatment (**Left panel**). The exogenous 27HC increased ROS production in BMECs (**Right panel**). (**B**) FACS plot showing the peIF2 expression level of the BMEC population after 27HC treatment (**Left panel**). peIF2 expression was increased in 27HC-treated BMECs (**Right panel**). Data represent the mean ± SEM, one-way ANOVA followed by Tukey’s multiple comparisons test. *** *p* < 0.001.

**Figure 6 ijms-25-10517-f006:**
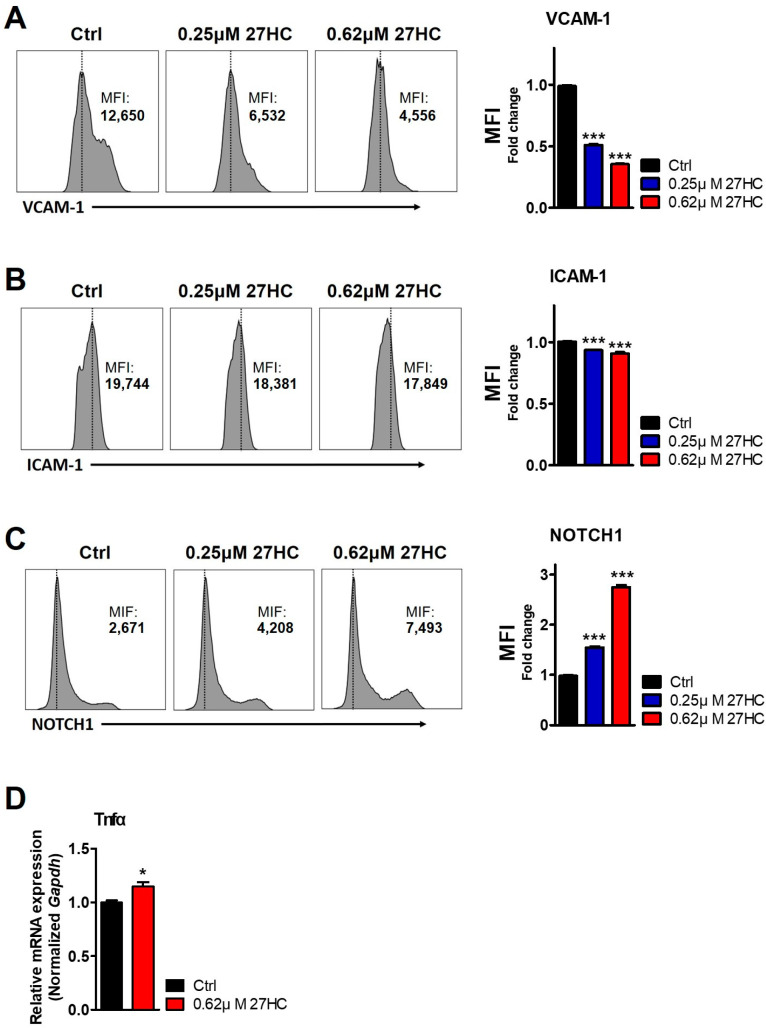
27HC induces BMEC dysfunction. CD31^+^ cells are treated with 27HC (0.25 µM and 0.62 µM) for 48 h. (**A**) FACS plot showing the VCAM1 expression level of the BMEC population after 27HC treatment (**Left panel**). The exogenous 27HC decreased VCAM1 expression in BMECs (**Right panel**). (**B**) FACS plot showing the ICAM1 expression level of the BMEC population after 27HC treatment (**Left panel**). ICAM1 expression was decreased in 27HC-treated BMECs (**Right panel**). (**C**) FACS plot showing the NOTCH1 expression level of the BMEC population after 27HC treatment (**Left panel**). NOTCH1 expression was increased in 27HC-treated BMECs (**Right panel**). (**D**) CD31^+^ cells were treated with 0.62 µM 27HC for 48 h. The relative levels of *Tnfα* mRNA were assessed using qRT-PCR. *Tnfα* expression was increased in 27HC-treated BMECs. Data represent the mean ± SEM, One-way ANOVA followed by Tukey’s multiple comparisons test. * *p* < 0.05, *** *p* < 0.001.

**Figure 7 ijms-25-10517-f007:**
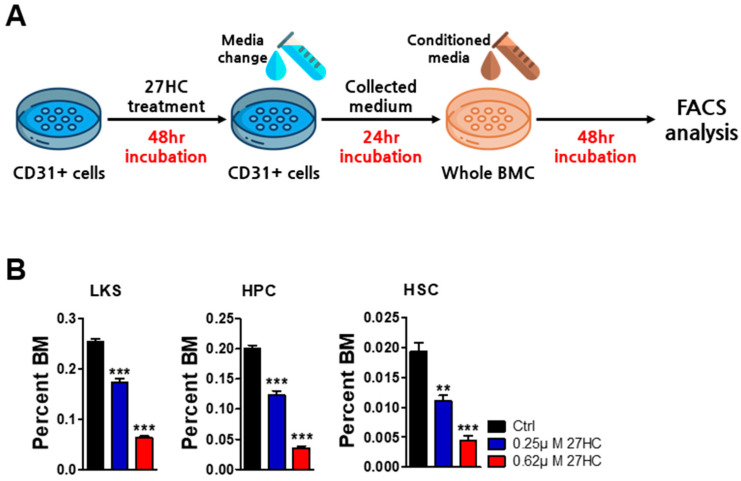
27HC-treated CD31^+^ BMEC conditioned medium reduces HSPCs. (**A**) Study overview. CD31^+^ cells isolated by MACS were treated with 27HC for 48 h and then changed to a growth medium without 27HC and incubated for 24 h. After 24 h, this conditioned medium was collected and used to treat mouse whole bone marrow cells (whole BMCs). The BMCs were collected and analyzed for the frequency of HSPCs. (**B**) Frequencies of LKS (Lin^−^Sca1^+^cKit^+^), HPCs (Lin^−^Sca1^+^cKit^+^CD48^+^), and HSCs (Lin^−^Sca1^+^cKit^+^CD150^+^CD48^−^) were significantly decreased by 27HC-treated CD31^+^ BMEC conditioned medium. Data represent the mean ±SEM, one-way ANOVA followed by Tukey’s multiple comparisons test. ** *p* < 0.01, *** *p* < 0.001.

**Table 1 ijms-25-10517-t001:** Antibodies for flow cytometric analysis.

Antibody	Company	Cat
PE-Cy5-CD3 (145-2C11)	BD, Franklin Lakes, NJ, USA	555276
PE-Cy5-CD4 (RM4-5)	BD	553050
PE-Cy5-CD8 (53-6.7)	Biolegend, San Diego, CA, USA	100710
PE-Cy5-CD19 (6D5)	Biolegend	115510
PE-Cy5-B220R (RA3-6B2)	eBioscience, San Diego, CA, USA	15-0452-82
PE-Cy5-Gr1 (RB6-8C5)	eBioscience	15-5931-82
PE-Cy5-Ter119 (TER119)	eBioscience	15-5921-82
PE-Sca1 (D7)	eBioscience	12-5981-82
APC-cKit (2B8)	Biolegend	105812
PE-Cy7-CD150 (Tc15-12F122)	Biolegend	115914
APC-Cy7-CD48 (HM48-1)	BD	561242
PE-CD31 (MEC 13.3)	BD	553373
APC-Cy7-Ter119 (TER119)	BD	560509
APC-Cy7-CD45 (30-F11)	Biolegend	103116
FITC-Sca1 (D7)	eBioscience	11-5981-82

**Table 2 ijms-25-10517-t002:** Primer sequences for flow quantitative RT-PCR.

Primer	Sequence	
mTNFα	Forward	5′ CTG AGG TCA ATC TGC CCA AGT AC 3′
	Reverse	5′ CTT CAC AGA GCA ATG ACT CCA AAG 3′

## Data Availability

The data supporting the findings of this study are available upon request from the corresponding author.

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
