# Peer review of "27-Hydroxycholesterol Negatively Affects the Function of Bone Marrow Endothelial Cells in the Bone Marrow"

_ijms, 2024, doi:10.3390/ijms251910517_

Round 1

Reviewer 1 Report

Comments and Suggestions for Authors

The manuscript investigated the effects of 27HC in BMEC abundance and function. The results showed that 27HC treatment in vitro and in vivo decreased the BMEC proportion and its expression of adhesion molecules ICAM1, VCAM1.  It can be improved by addressing the following concerns.

1.       The statistical analysis has problem. Unpaired t-test is suitable for comparing two groups with equal variation. In figure 1, 2, 4, and 5 have three or more groups, One-way ANOVA with multiple group comparison is suggested to be used with equal variation.

2.       The statements throughout the manuscript need to be rewording. Some statements are vague and not specific. For example, the authors conclude that 27HC impaired BMECs with only the cell percentage result. Some statements are not accurate. For example, the result showed treatment of 1.24uM 27HC decreased HSCs, but the authors claims that 27HC does not affects HSCs. Some statements are overly stated. For example, the authors conclude that the presence of BMECs is crucial for the survival of HSCs while not data are supported with this statement.

3.       To support the conclusion that 27HC impaired adhesion molecular expression in BMECs, ICAM1 and VCAM1 mRNA and protein levels should be detected after 27HC treatment. Mechanistic studies, such as pathway analysis using WB, are suggested to proceed.

Reviewer 2 Report

Comments and Suggestions for Authors

The manuscript explores the impact of 27-hydroxycholesterol (27HC) on bone marrow endothelial cells (BMECs) and hematopoietic stem cells (HSCs), with a focus on how 27HC modulates BMEC function and consequently influences HSC survival. The research is compelling and addresses an important gap in understanding how cholesterol metabolites, particularly 27HC, regulate hematopoietic niches. The study is well-structured, with clear experimental designs and a logical flow from hypothesis to conclusion. However, there are several aspects where the manuscript could be improved for clarity, depth of analysis, and broader applicability.

Comments:

1. While the study demonstrates that 27HC negatively impacts BMECs and their ability to maintain HSCs, the mechanistic pathways through which 27HC exerts its effects are not sufficiently explored. The authors touch upon increased reactive oxygen species (ROS) and endoplasmic reticulum (ER) stress as potential mediators but do not provide experimental data to support this in BMECs. Further exploration of these pathways could enhance the mechanistic depth of the study.

2. The manuscript suggests that 27HC primarily affects BMECs and not HSCs. However, the reduction in adhesion molecules such as ICAM1 and VCAM1 suggests that the long-term maintenance of HSCs could be compromised, even if short-term effects are not visible. The manuscript does not explore whether longer exposure to 27HC would impact HSC function indirectly through BMEC dysfunction

3. The manuscript focuses on the specific effects of 27HC but does not sufficiently discuss the broader implications of these findings. For instance, hypercholesterolemia is a common condition, and its potential to affect hematopoiesis via BMEC dysfunction is an important public health concern.

4. The manuscript acknowledges that BMECs are heterogeneous but focuses primarily on sinusoidal and arteriolar BMECs. Other subtypes, such as capillary BMECs, are mentioned but not explored in depth. This limits the generalizability of the findings to the full spectrum of BMECs.

5. Some figures, particularly those representing flow cytometric data, are difficult to interpret due to unclear labelling and resolution issues. Figure legends should provide more context to guide the reader through the results presented

Reviewer 3 Report

Comments and Suggestions for Authors

In the manuscript „27-Hydroxycholesterol negatively affects the function of bone marrow endothelial cells in the bone marrow” authors report on the effect of 27HC on BMEC which are important in maintaining normal hematopoiesis. 

Broad comments: Manuscript is difficult to read so English should be thoroughly corrected. The manuscript reports on the interesting matter, but the main drawback it the lack in mechanistic data behind the effects observed. 

Major:

1. What is the mechanism behind the difference in the response to 27HC between HSC and BMEC? The authors should measure ROS in BMEC and compare it to HSC, as well as ER stress.

2. The authors should perform the mechanistic analysis by investigating NOTCH signaling (WB, PCR) in BMEC in response to 27HC. Furthermore, an inhibition on NOTCH signaling and its effects on BMEC response should be investigated.

3. It would be interesting to see the concentration of the cytokines in the medium of CD31+ cells in response to 27HC

Minor: 

1. Line 36 – correct english „led to dose-dependently decreased”

2. Line 53 – correct English produce “growth and inhibitory cytokines…”

3. Line 55 – explain what KITL is

4. Line 97-99 – reference for previous studies

5. Figure 1 – time point should be both in the Figure and in the figure legend

6. Figure 1 – what was the initial count/density of seeding bm cells. Which medium was used?

7. Figure 2 – figures should be self-explanatory, provide which chemical was used and incubation time in the figure as well

8. Why did you use the concentration of 6.2 µ M for 27HC, reference on this?

9. Introduction – provide information which studies were performed in mice, which in humans, which were in vitro etc…

10. Add in the beginning of results section that it was isolated murine bone marrow   

11. Line 104 – there is no Figure 2C

12. Line 126 – again why these concentrations

13. Line 129-130 – this is unclear, in the figure you have LKS, HPC, HSC but in the text you state different. Additionally, explain each.   “, long-term 27HC administration did not affect the number of LKS (Lin-Sca1+cKit+ ), hematopoietic stem cells (Lin- 130 Sca1+cKit+CD48* ), and HSCs”.

14. Line 149 – explain CD31+ and CD31-  cells and why you performed these additional experiments

15. Line 161-162 – It is difficult to understand this sentence -   The frequencies of BMECs from CD31+ cells and HSCs from CD31- cells were analyzed using flow cytometric analysis

16. Figure 4 – why only low-dose 27HC?

17. Line 174 – full names should have been mentioned earlier in the manuscript

Comments on the Quality of English Language

Extensive editing of English language required.

Round 2

Reviewer 1 Report

Comments and Suggestions for Authors

The authors have addressed all my concerns.

Reviewer 3 Report

Comments and Suggestions for Authors

No further revisions needed.